# Sphingosine 1-Phosphate as Essential Signaling Molecule in Inflammatory Skin Diseases

**DOI:** 10.3390/ijms24021456

**Published:** 2023-01-11

**Authors:** Burkhard Kleuser, Wolfgang Bäumer

**Affiliations:** 1Department of Pharmacology and Toxicology, Institute of Pharmacy, Freie Universität Berlin, Königin-Luise Str. 2+4, 14195 Berlin, Germany; 2Department of Veterinary Medicine, Institute of Pharmacology and Toxicology, Freie Universität Berlin, Koserstr. 20, 14195 Berlin, Germany

**Keywords:** sphingolipids, ceramides, keratinocytes, fibroblasts, T-cells, dendritic cells, mast cells, atopic dermatitis, psoriasis, itch

## Abstract

Sphingolipids are crucial molecules of the mammalian epidermis. The formation of skin-specific ceramides contributes to the formation of lipid lamellae, which are important for the protection of the epidermis from excessive water loss and protect the skin from the invasion of pathogens and the penetration of xenobiotics. In addition to being structural constituents of the epidermal layer, sphingolipids are also key signaling molecules that participate in the regulation of epidermal cells and the immune cells of the skin. While the importance of ceramides with regard to the proliferation and differentiation of skin cells has been known for a long time, it has emerged in recent years that the sphingolipid sphingosine 1-phosphate (S1P) is also involved in processes such as the proliferation and differentiation of keratinocytes. In addition, the immunomodulatory role of this sphingolipid species is becoming increasingly apparent. This is significant as S1P mediates a variety of its actions via G-protein coupled receptors. It is, therefore, not surprising that dysregulation in the signaling pathways of S1P is involved in the pathophysiological conditions of skin diseases. In the present review, the importance of S1P in skin cells, as well as the immune cells of the skin, is elaborated. In particular, the role of the molecule in inflammatory skin diseases will be discussed. This is important because interfering with S1P signaling pathways may represent an innovative option for the treatment of inflammatory skin diseases.

## 1. Introduction

### 1.1. Sphingolipids as Essential Structural Components of the Skin

In recent years, it has been discovered that sphingolipids, such as ceramides and sphingosine 1-phosphate (S1P), are important signaling molecules in the skin. Before that, it was assumed that sphingolipids were primarily responsible for building up the epidermal barrier. It is all the more surprising that specific sphingolipid species are also involved in the regulation of the epidermal barrier. Since the epidermal barrier plays such an important role, the structural composition, with a focus on skin-specific sphingolipids, is briefly described.

The barrier between the environment and the human organism is formed by the stratum corneum, the outermost layer of the skin. The stratum corneum provides a primary layer of protection against countless increasingly damaging invaders and chemicals and reduces transepidermal water loss (TEWL) [1]. Normally, the stratum corneum consists of 15–25 layers of terminally differentiated keratinocytes, called corneocytes, embedded in an extracellular lipid environment. These lipid lamellae are composed of sphingolipids, cholesterol, and free fatty acids in an approximately equimolar ratio and are arranged in a dense orthorhombic lateral packaging in the interstices of the stratum corneum. Ceramides are considered to be one of the most important epidermal sphingolipids and represent approximately 50% of the intercellular lipids of the stratum corneum [2,3]. Ceramides are composed of a sphingoid base backbone linked to a fatty acid via an amide bond. At least four types of sphingoid bases, namely sphingosine [S], 6-hydroxysphingosine [H], dihydrosphingosine [DS], and phytosphingosine [P], occur in the stratum corneum, differing in the degree of unsaturation and the position and number of the hydroxyl groups (Figure 1). Different types of fatty acids, esterified ω-hydroxy [EO], ω-hydroxy [O], α-hydroxy [A], and non-hydroxy [N] fatty acids, presented in Figure 1, are proven to occur at the sphingoid bases, leading to a multitude of different ceramide species [4]. The orthorhombic lateral packing of these ceramide species is essential in maintaining the skin’s lipid lamellae structure.

Skin barrier formation is a fine-tuned procedure in which proliferating basal keratinocytes cease mitotic activity, differentiate, and migrate through four epidermal layers (stratum basale, stratum spinosum, stratum granulosum, and stratum corneum), thereby adjusting to the demands of the particular cell layer. It has been indicated that three routes contribute to the generation of ceramides in the skin [5]. The de novo pathway, which is presented in Figure 2, takes place at the outer membrane of the endoplasmic reticulum and is initiated by the action of serine palmitoyltransferase starting with the condensation of the preferred substrates serine and palmitoyl-CoA. The resultant 3-ketosphinganine is then reduced by 3-ketosphinganine reductase to sphinganine. Ceramide synthases link fatty acyl CoAs to the amino group of sphinganine, leading to the formation of dihydroceramides. The introduction of a double bond between carbons C-4 and C-5, mediated by dihydroceramide desaturase, forms the central product of ceramide. It is a peculiarity of keratinocytes that the enzymes that lead to the formation of skin-specific acylated ceramides are also localized in the endoplasmic reticulum. The fatty acid elongases ELOVL1 and ELOVL4 yield ultralong-chain fatty acids of up to C26 and C28 carbon-chain-lengths, respectively [6,7]. The cytochrome P450 enzyme CYP4F22 subsequently hydroxylates at the ω-position, generating ω-hydroxylated ultralong-chain fatty acids [8]. Ceramide synthase 3 uses these ω-hydroxylated fatty acids for ceramide formation [9]. Finally, the transacylase PNPLA1 (Patatin-like Phospholipase Domain Containing 1) generates an ester linkage between the fatty acid taken from triglycerides and the ω-hydroxy group from ceramide to create such an acylceramide [10].

However, ceramides can also be generated via the degradation of glucosylceramides by β-glucocerebrosidase and the hydrolysis of sphingomyelin by sphingomyelinase [11]. It has been indicated that excessive keratinocyte-specific ceramide synthesis is already increased in the suprabasal cell layer via the de novo pathway. These ceramides are transported from the endoplasmic reticulum to the Golgi apparatus, where the conversion to glucosylceramides or sphingomyelin occurs. These lipids are then encapsulated into epidermal lamellar bodies, exclusive secretory organelles of keratinocytes, which become more abundant with the differentiation state of keratinocytes. Lamellar bodies secrete glucosylceramides and sphingomyelins into the extracellular environment at the interface of the stratum granulosum and the stratum corneum. There they are reconverted back to their ceramide species due to an activation of the enzymes ß-glucocerebrosidase and sphingomyelinase, forming repeated sheets of lipid lamellae in the stratum corneum [12,13,14,15]. Moreover, some acylceramides at the fatty acid position are subjected to peroxidation via the lipoxygenases ALOX12B and ALOXE3, followed by deglycosylation. These ω-hydroxy ceramides can then covalently bind to the protein structures of the corneocytes, such as involucrin and envoplakin, leading to a stable lipid–cell scaffold (Figure 2).

### 1.2. Sphingolipids as Signaling Molecules in the Skin

Beyond their structural properties, sphingolipids have been identified as crucial compounds that are also involved as signaling molecules during barrier formation as they communicate with the mitotic active keratinocytes residing in the stratum basale. Thus, even the involvement of ceramides in the essential functions of keratinocytes has been well documented. In cultured keratinocytes, ceramides have been shown to diminish cell proliferation and increase cell differentiation [16,17]. This is in congruence with the signaling pathways such as protein kinase B (Akt), protein kinase C (PKC), mitogen-activated protein kinase (MAPK), Jun N-terminal kinase (JNK), or phospholipase D (PLD) that are modulated in response to ceramide stimulation [18]. Thus, elevated ceramide levels in differentiating keratinocytes are not only required for lamellar membrane production but also corneocyte development.

It has also been known for quite some time that another bioactive sphingolipid, namely S1P, is involved in the modulation of keratinocyte-specific actions [19]. Noteworthy, S1P influences not only keratinocytes but also the immune cells located in the skin, indicating that S1P, therefore, plays a major role in skin inflammatory processes. The present review summarizes the importance of this sphingolipid species on different skin cell types; in particular, a distinction is also made here from the systemic effects that emanate from S1P and which can be modulated by the systemic action of S1P-receptor modulators.

## 2. Sphingosine 1-Phosphate Metabolism, Signaling, and Immune Cell Migration

### 2.1. Sphingosine 1-Phosphate Metabolism

The bioactive sphingolipid, S1P, occurs in tissues at relatively low concentrations owing to its ability to exert its effects via receptor-mediated actions. However, intracellular target proteins have also been identified to be directly modulated via sphingolipids increasing the complexity of S1P signaling. In the sphingolipid metabolic pathways, ceramide serves as a crucial nexus for the generation of S1P as it is the only known precursor for the formation of sphingosine from which S1P is generated. Sphingosine is formed via the deacylation of ceramides via ceramidases, which is then phosphorylated by either sphingosine kinase 1 (SPHK1) or 2 (SPHK2), thereby producing the bioactive molecule S1P [20]. S1P is able to modulate a multitude of cellular processes, such as proliferation and migration, the regulation of epigenetic key enzymes, angiogenesis, and lymphangiogenesis [21]. Consequently, S1P biosynthesis and breakdown are precisely balanced, and disequilibrium in such enzymes that control these processes can lead to pathological conditions, including defective skin barrier formation (Figure 3).

The functions of the two SPHK enzymes appear to overlap in some respects, as the selective knockdown of SPHK1 or SPHK2 does not result in any vital deficits. However, the simultaneous deletion of SPHK1 and 2 results in embryonic lethality. In fact, SPHK1 and SPHK2 often possess divergent functions and even have contrasting roles in cellular processes depending on their distinctive tissue occurrence, cellular compartmentalization, and cellular regulation [22]. SPHK1 is typically found in the cytoplasm, and the enzyme can be stimulated by phosphorylation through extracellular signal-regulated kinases (ERK 1/2) or other signaling bioactive lipids, such as ceramide 1-phosphate [23]. The stimulation of SPHK1 is accompanied by translocation to the plasma membrane, a process where calcium- and integrin-binding proteins are involved [24]. Nevertheless, an additional modulation of SPHK1 occurs during the transcriptional stage [25]. In contrast, SPHK2 is the less closely studied of the two isoenzymes. It has been indicated that SPHK2 is also activated via phosphorylation in response to ERK1/2. Concerning its localization, SPHK2 is mainly found in the nucleus and at the inner mitochondrial membrane, which enables the specific crosstalk of S1P with nuclear and mitochondrial proteins and enzymes [26]. The elimination of S1P takes place due to the irreversible breakdown of the sphingosine backbone at the C2-C3 position, resulting in the generation of hexadecenal and phosphoethanolamine. This process is catalyzed by S1P lyase. S1P-lyase knockout mice die a few weeks after weaning. These mice show several severe developmental abnormalities and a pronounced dysregulation of lipid species in various organs, such as the liver and brain [27]. The deletion of S1P lyase is accompanied by an enhanced sphingolipid generation via the salvage pathway, whereas the de novo pathway is decreased [28]. In addition to the irreversible degradation, S1P can also be reversibly dephosphorylated to sphingosine via the action of two specific isoforms of the S1P phosphatases SGPP1 and SGPP2, as well as via the action of nonspecific lipid phosphate phosphatases (LPP1-3) [29].

Since S1P is formed within the cell and a variety of effects are conveyed by the stimulation of S1P receptors (S1PR) in a paracrine and/or autocrine manner, a dedicated shuttle for the transport of S1P into the extracellular matrix must exist. Indeed, a number of transport proteins have been found to shuttle S1P across the cell membrane. Spinster homologue 2 (SPNS2), a member of the superfamily of non-ATP-dependent organic ion transporters, is a specific transporter for S1P in a variety of cells [30]. However, S1P can also be transported via the members of the ATP-binding cassette (ABC) transporters, a protein family that plays a central role, especially in keratinocytes [31].

### 2.2. Sphingosine 1-Phosphate Signaling

Generated S1P is able to activate five S1P-specific G-protein-coupled receptors (S1PR1–5), which are cell-specific expressed and interact with a wide array of heterotrimeric G proteins, thereby leading to broad, and sometimes contradictory cellular and physiological effects [32]. The S1P generated inside the cell also has the ability to regulate intracellular responses; nevertheless, in distinction to the receptor-mediated effects, only a limited number of intracellular targets have been identified so far. Thus, an important intracellular binding partner of S1P is tumor necrosis factor (TNF) receptor-associated factor 2 (TRAF2), a key adaptor protein in TNF-receptor signaling. The binding of S1P to TRAF2 in response to the stimulation of SPHK1 promotes the TNF-receptor signaling downstream pathways, resulting in the activation of the major pro-inflammatory transcription factor, NF-κB [33]. Another intracellular target of S1P is connected to epigenetic regulation. S1P, which is generated via SPHK2, is able to inhibit histone deacetylases (HDAC), a class of enzymes that remove acetyl groups from an *N*-acetyl lysine amino acid on a histone, thus allowing the histones to wrap around the DNA more tightly. This is of interest, as HDAC inhibition is connected to cancer-cell-cycle arrest, differentiation, and cell death [34]. On the contrary, it has been shown that intranuclearly generated S1P also binds to telomerase reverse transcriptase and increases telomerase activity, which may enhance cancer cell growth [35].

### 2.3. Sphingosine 1-Phosphate and Immune Cell Migration

In recent years it has been well established that S1P plays a fundamental role in immune cell trafficking. One function of S1P is to regulate the egress of lymphocytes from the lymph nodes into the bloodstream, from where they can target the site of inflammation. A finely tuned S1P gradient between the lymph nodes, lymph, and blood is essential for this action. Normally, mature lymphocytes in search of antigens circulate between secondary lymphoid organs, such as lymph nodes, where antigens from local tissues are present [36]. Lymphocytes reach the lymph nodes from the circulation via high endothelial venules. A lymphocyte spends some time scanning the lymph node for antigens, and if it detects none, it leaves the lymph node via the lymph and returns to the blood, and the process starts all over again. However, when the lymphocyte hits its cognate antigen, it will remain in the lymph node, proliferate, and gain an effector role. An important component of gaining effector function is modifying the expression of homing receptors, leading to an optimal migration of the lymphocyte to reach the site of the infection. The crucial role of S1P receptors in the process of lymph-node-exit signaling was first demonstrated by studies using FTY720 [37].

FTY720, also known as Fingolimod, was the first drug approved by the US Food and Drug Administration (FDA) for the treatment of relapsing–remitting multiple sclerosis, a disorder in which T cells contribute to the damage of the myelin layer of neurons [38]. Indeed, FTY720 was shown to be able to prevent T cells from leaving the lymph nodes, and the molecular mechanism of how FTY720 exerts its effect has also been elucidated [39,40]. FTY720 is a prodrug that must first be phosphorylated via SPHK2 in vivo to its active compound FTY720-phosphate (FTY720-P). As an S1P analog, FTY720-P binds to four of the five S1P receptor subtypes, and only the S1PR2 receptor is not targeted. FTY720-P has mixed agonist–antagonist effects on S1PR1. Initially, there is a strong activation of this receptor subtype. It should be mentioned that this is also the cause of the cardiac side effects of bradycardia that can occur at the beginning of therapy [41]. However, the inactivation of S1PR1 occurs during the further response to FTY720-P stimulation. This is due to the fact that FTY720-P brings S1PR1 into a conformation that leads to preferential ubiquitination and degradation rather than recycling [42]. The result is a decreased number of S1PR1s on the cell surface. In accordance, it has been shown by the use of a mouse model with an internalization-resistant S1PR1 that the ability of FTY720 to inhibit lymphocyte egress is significantly diminished [43]. Moreover, that this S1PR1 is responsible for the egress of lymphocytes from the lymph node has also been shown in S1PR1-deficient T and B cells, as they are able to enter the lymph node but are unable to exit it [39]. It is of interest that S1PR1 signaling in T cells competes with the retention signals from CCR7 and CXCR4. This suggests that S1PR1 is essential in controlling exit, especially in the presence of retention signals [44].

Because S1PR1 expression and other retention signals are dynamically regulated during an immune response, a variety of S1P receptor modulators other than FTY720 have been developed and tested in clinical trials in several autoimmune diseases, such as psoriasis [45].

However, in addition to the central role of S1P in the circulation of T cells, it has also emerged that S1P, and its receptor-mediated signaling, especially in addition to S1PR1, affects almost every cell type, including the cells of the skin. Therefore, in the following, the effects of S1P on the skin cells and immune cells present in the skin will be discussed. The focus is also on the fact whether a topical application of S1P or S1P modulators could play an important role in inflammatory skin diseases.

## 3. Sphingosine 1-Phosphate and Skin Cells

### 3.1. Sphingosine 1-Phosphate and Keratinocytes

Compared to many cell types, S1P has a unique feature in keratinocytes. This is because S1P promotes proliferation in many cells, but an opposite effect is visible in keratinocytes, where proliferation is inhibited. The treatment of keratinocytes with S1P is accompanied by the inhibition of cyclin D synthesis and the stimulation of p21(WAF1/CIP1) (p21) and p27(KIP1) (p27) synthesis. As a consequence, cyclin-dependent kinase is inhibited, and the cell cycle is arrested at the G1 phase [46,47,48]. Thus, with respect to cell growth, S1P has analogous effects to ceramides, but the two molecules differ in their modulation of apoptosis. While ceramides possess a proapoptotic effect, S1P protects keratinocytes from programmed cell death [47]. Moreover, S1P enhances the differentiation of keratinocytes. The typical differentiation markers of early and late differentiation, such as keratin 1 and involucrin, are enhanced in response to S1P stimulation [48]. Regarding the inhibitory effect on cell growth, it has been clearly demonstrated that the prolonged activation of ERK and the transient inactivation of Akt is the crucial pathway in S1P-mediated cell-growth arrest [46]. Calcium is required to initiate the differentiation of keratinocytes as it upregulates the genes involved in the differentiation process. Indeed, S1P causes transient increases in intracellular free Ca^2+^ concentrations. This enhancement is obviously mediated by the stimulation of phospholipase C and involves Ca^2+^ mobilization from thapsigargin-sensitive stores and subsequent Ca^2+^ influx [49]. The molecular mechanism of the anti-apoptotic action induced by S1P in keratinocytes is less characterized. However, it has been shown that S1P induces the activation of endothelial nitric oxide synthase (eNOS) in human keratinocytes, leading to a moderate enhancement of nitric oxide. The cell-protective effect of S1P is diminished in eNOS-deficient keratinocytes, indicating that S1P protects keratinocytes from apoptosis via eNOS activation [50]. This is consistent with the fact that autologous nitric oxide protects human keratinocytes from ultraviolet-B-radiation-induced apoptosis [51].

The effects on proliferation and differentiation are also apparent when the enzymes that affect the S1P levels are modulated. Thus, intracellular S1P levels are increased in S1P phosphatase 1-deficient keratinocytes. This is visible in the keratinocytes isolated from the skin of SGPP1-deficient pups. The increased intracellular S1P levels are accompanied by a gene expression profile that indicates the overexpression of the genes associated with keratinocyte differentiation [52]. A similar result can be obtained through the inhibition of S1P lyase, which leads not only to a cell cycle arrest but also to the upregulation of the differentiation markers of keratinocytes [53].

The results have to be considered in a slightly more discriminating way when both SPHK1/2 are modulated. The hydrophobic compound K6PC-5 has been identified as an activator of SPHK1. The treatment of keratinocytes with K6PC-5 inhibits their proliferation and increases the expression of the differentiation markers involucrin and filaggrin. In congruence with an S1P treatment, K6PC-5 also induces an influx of intracellular Ca^2+^ concentrations. These effects are abolished when SPHK1 is abrogated via siRNA, indicating that K6PC-5 acts to regulate both the differentiation and proliferation of keratinocytes via S1P production [54].

This contrasts with the results obtained when the inhibition of SPHK2 occurs. HWG-35D and ABC294640 represent inhibitors of SPHK2, and topical treatment with these inhibitors reduces the expression of keratin K6 and K16 of stressed keratinocytes at the suprabasal layers of the epidermis [55,56]. As keratin K6 and K16 are markers of keratinocyte hyperproliferation, these results indicate that the inhibition of SPHK2 contributes to an antiproliferative effect in keratinocytes. However, at least, ABC294640 also affects further enzymes of the sphingolipid pathway, such as dihydroceramide desaturase [57,58].

In keratinocytes, all five S1P receptor subtypes are expressed, and the subtypes of S1PR2 and S1PR3 appear to be particularly responsible for the S1P-mediated effects on proliferation, differentiation, and cytoprotection. Thus, the putative S1PR3 antagonist, BML-241, inhibits the S1P-induced Ca^2+^ increase, the crucial event for differentiation [49]. Additionally, S1P almost completely loses its ability to protect human keratinocytes from apoptosis when the S1PR3 is abrogated [50]. Regarding cell growth arrest, S1PR2 plays an important role. The abrogation of S1PR2 restores not only the inhibitory effect of S1P on Akt phosphorylation but also prevents S1P-induced growth arrest [59]. In agreement, FTY720-P, not acting on the S1PR2, is not able to inhibit the cell growth of keratinocytes. S1PR2 have also been identified as a crucial receptor for the maintenance of epidermal barrier homeostasis. A recent publication indicated that S1PR2 knockout mice have more fragile skin due to the reduced expression of tight junction proteins, such as Zo-1, occludin, claudin-1, and cornedesmosin in the keratinocytes of the epidermis of S1PR2-deficient mice [60]. As a consequence, these mice show a slightly enhanced TEWL as a measurement for the barrier defect, which increases dramatically upon tape stripping. In addition, filaggrin 2 expression is also negatively regulated in S1PR2 knockout mice. Consequently, the topical administration of Staphylococcus aureus led to the deeper penetration of bacteria and more severe signs of infection in S1PR2 knockout mice.

### 3.2. Sphingosine 1-Phosphate and Dendritic Cells

The skin contains a variety of specialized antigen-presenting cells (APCs) that belong to the family of classical dendritic cells (DCs). A crucial DC subtype in the skin are Langerhans cells (LCs), which form dense cellular networks in the basal and suprabasal layers of the epidermis [61,62]. They sense exogenous molecules that have penetrated the skin barrier and relay this information to the skin’s lymph nodes. LCs exhibit a distinct capacity for phagocytosis and are characterized by intracellular organelles, Birbeck granules, which act as endosomal recycling spaces. The antigen-processing activity of Birbeck granules is linked to the surface expression of the endocytotic receptor Langerin (CD207). A further subtype of DCs in the skin are dermal DCs (dDCs), which are located in the dermis and also participate in antigen presentation. Thus, the function of LCs and dDCs is the uptake, endosomal processing, and presentation of antigens, which endows them with the unique ability to elicit adaptive immune responses and induce and control tolerance. When immature LCs/dDCs capture and process antigens, they migrate toward secondary lymphoid organs, where they interact with naive T cells. As they migrate, the capacity of LCs/dDCs to take up additional antigens diminishes, and they acquire the competence to present antigens to naïve T cells in a properly synchronized sequence of events known as maturation.

S1P interferes with numerous processes of the physiological functions of LCs/DCs, including migration, antigen uptake, and maturation. With respect to antigen capture, a concentration-dependent reduction in endocytotic capacity in immature LCs/DCs by S1P could be determined. Several endocytosis mechanisms are available for the uptake of antigens by immature LCs/DCs from their immediate environment, which can be subdivided into macropinocytosis, phagocytosis, and receptor-mediated endocytosis. In fact, S1P has been shown to reduce macropinocytosis, the unspecific fluid uptake and the antigens dissolved therein. The investigation of the signaling pathway responsible for this showed the regulation of phosphoinositide-3-kinase (PI3K) activity as the cause of the altered endocytosis behavior of immature LCs/DCs. Moreover, the stimulation of SPHK activity is connected not only with enhanced S1P levels but also with diminished antigen capture. In congruence, the decreasing activity of PI3K involved in actin polymerization or macropinocytosis is accompanied by increasing S1P concentrations. Most interestingly, S1PR2 has been identified as the crucial receptor subtype to inhibit this antigen capture [63].

However, another effect has also been demonstrated for the S1PR2 subtype in DCs. S1PR2 appears to be involved in the expression of CCL17 and CCL22 in mature DCs [64]. This is of interest, as CCL17 and CCL22 are known to assist lymphocytes in targeting their response to skin-located pathogens. They accomplish this by binding to CCR4, a chemokine receptor expressed especially on type 2 helper T cells, which then release a multitude of cytokines.

There are further reports that S1P directly influences cytokine release in DCs. Pro-inflammatory cytokines of the IL-12 family play a central role in the development and maintenance of inflammatory diseases. The IL-12 family is unique in that it comprises the only heterodimeric cytokines, which include IL-12 (p40/p35), IL-23 (p40/p19), IL-27 (EBI3/p28), and IL-35 (p35/EBI3) [26,65]. Although they share many structural features and molecular partners, they have surprisingly different functional effects. It has been indicated that S1P modulates IL-12 production in human DCs. Mature DCs show a diminished secretion of IL-12 and TNF-α but an enhanced release of IL-10. The suppression of IL-12 production in DCs in response to S1P is accompanied by a Th1 to Th2 switch of in vitro primed T cells [66]. It has been demonstrated that SPHK1-delivered extracellular S1P inhibits the production of IL-12 in DCs via S1PR1 [67].

However, there have been no studies on the cytokines IL-23 and IL-27. To further characterize the influence of S1P on cytokine production, murine bone marrow derived DCs were utilized. S1P significantly reduced IL-12 and IL-23 production induced by LPS, while the secretion of IL-27 was not inhibited. Since IL-12 and IL-23 have a common subunit (p40) in contrast to IL-27, it could be demonstrated that S1P inhibits IL-12/23p40 secretion in a concentration-dependent manner [65].

S1P and S1P-regulating enzymes have also been indicated to modulate the function of plasmacytoid dendritic cells (pDCs), also known as natural interferon (IFN)-producing cells. These cells comprise a specific cell type within the innate immune system. pDCs are able to recognize viral RNA and DNA via Toll-like receptor (TLR)-7 and TLR-9. Consequently, they rapidly secrete large amounts of type 1 IFN upon contact with the virus. Then, pDCs start to differentiate into professional APCs, which possess the capacity to stimulate the T cells of the adaptive immune system. pDCs can be found in primary lymphoid organs and the lymph nodes of secondary lymphoid tissues. While generally lacking in normal skin, pDCs infiltrate the skin in several pathologies, such as psoriasis and systemic lupus erythematosus. It has been shown that treatment of pDCs with S1P decreases IFN-α production via S1PR4. Thus, S1PR4 signaling inhibits the IFN-α production by a diminished internalization of the pDC-specific inhibitory receptor, Ig-like transcript 7 [68]. In addition, S1PR1 seems to be involved in IFN-α production. The stimulation of S1PR1 in pDCs diminishes IFN autoamplification through the induced degradation of the IFN-α receptor 1 and the suppression of STAT1 signaling [69]. The reduced IFN-α production via S1P also affects the production of cytokines in a coculture system with T cells from a Th1 (mainly IFN-γ) to a regulatory profile characterized by enhanced IL-10 production [68]. This implicates a beneficial role of S1P, particularly in the context of psoriasis, where pDC are found to be increased in lesional skin and where a Th1/Th17 cytokine profile is dominant [70]. However, a controversial role has been implicated concerning the role of SPHK1. The pharmacological inhibition or genetic deletion of SPHK1 results in a decreased production of type I IFN. It has been suggested that intracellular S1P is essential for the efficient uptake of TLR7/9 ligands and trafficking to endosomes and that this effect is independent of the extracellular actions of S1P as a ligand of S1PRs. Therefore, a clear distinction between the intracellular effects and receptor-mediated functions of S1P must also be made for pDC [71].

### 3.3. Sphingosine 1-Phosphate and Macrophages

Macrophages are innate immune cells that play an important role in the initiation of inflammatory processes but also participate in the resolution of inflammation. Depending on their functional role, pro-inflammatory “classically activated” (M1) macrophages (stimulus, e.g., LPS and IFN-γ) are distinguished from anti-inflammatory “alternatively activated” (M2) macrophages (stimulus, e.g., IL-4). It has been demonstrated that human macrophages express S1PR1-4 [72]. In contrast, murine macrophages seem to express particularly S1PR1 and 2 [73]. Comparable to DCs, S1P is essential for macrophage migration under physiological and pathophysiological conditions [74]. In an inflammatory condition (mouse peritonitis model), S1PR2 inhibited macrophage recruitment while S1PR3 promoted it [73,75]. The particular role of S1PR1 was evaluated in a separate study. Interestingly, a classic activation of macrophages with IFN-γ and LPS does not induce an upregulation of S1PR1, which would be a pre-requisite for the emigration of macrophages from the site of inflammation. However, alternatively activated macrophages (by IL-4) display upregulated S1PR1 and thus emigrate out of the site of inflammation during resolution. This was confirmed in an in vivo model of peritonitis in mice, where the enhanced migration of macrophages during resolution was abolished in S1PR1 knockout mice [76].

More recently, a novel function of S1PR4 on macrophage function was evaluated by the same group [77]. In the imiquimod-induced mouse model of psoriasis (see below), the PASI score was moderately reduced in S1PR4 knockout mice. This was accompanied by a reduced influx of macrophages into the inflamed skin. Again, a similar effect was observed in a peritonitis model, but with fewer macrophages at the site of inflammation. This was preceded by reduced levels of CCL2 and IL-6. Particularly CCL2 is a chemoattractant for macrophages, and at the same time, macrophages are a main source of CCL2. The incubation of macrophages with zymosan induced marked CCL2 expression in macrophages generated from wild-type mice but not in those generated from S1PR4 knockout mice [77]. Generally, the incubation of macrophages with S1P seems to have predominantly anti-inflammatory effects due to the significantly reduced production of the pro-inflammatory cytokines TNFα, IL-6, IL-12, and CCL2 after their activation by, e.g., LPS [78].

### 3.4. Sphingosine 1-Phosphate and Mast Cells

Mast cells are central players in allergic diseases, such as atopic dermatitis. Mast cells express high-affinity receptors for IgE, FcεRI. Via IgE-dependent and independent activation and degranulation, mast cells secrete lysosomal enzymes, proteases, and histamine. Antigen-stimulated mast cells also actively produce and secrete a wide variety of lipids and proteins that require de novo synthesis. Among de novo synthesized lipid mediators are eicosanoids, such as prostaglandins, leukotrienes, and S1P. The activation of FcεRI facilitates the recruitment of both SPHK1 and SPHK2 to lipid rafts in proximity to their substrate. A peak of S1P synthesis in mast cells upon FcεRI activation occurs within minutes. However, a second increase in S1P synthesis occurs after 30 min, and this is accompanied by enhanced S1P concentrations in the extracellular environment [79].

S1P excreted into the extracellular environment can act on mast cells in an autocrine manner. Mainly two receptors (S1PR1 and S1PR2) are described for mast cells [79]. Two pivotal mast cell functions, mast cell migration as well as degranulation, are modulated by S1P via binding to the extracellular receptors (intracellular signals via S1P are much less understood but also seem important).

SPHK activation and S1PR1 signaling are involved in the migration of mast cells toward antigens. Depending on the mast cell type and the experimental settings, S1PR2 can play a central role in FcεRI-induced degranulation [80]. However, in other settings, the role of S1PR2 seems less clear/pronounced [81]. Nevertheless, pharmacological inhibition of S1PR2, e.g., by JTE013, impairs degranulation and cytokine and chemokine secretion from human mast cells [82]. This is corroborated in vivo in a passive systemic anaphylaxis experiment. S1PR2 knockout mice or wild-type mice treated with the S1PR2 antagonist JTE013 show dramatically reduced signs of IgE-mediated anaphylaxis (drop in temperature, histamine release, and lung edema) [82]. Chronic exposure to high concentrations of S1P seems to change the mast cells’ phenotype. Incubation with S1P leads to a hyper-responsive phenotype [81]. However, the complex role of S1P is once more revealed by the fact that an injection of S1P can diminish the signs of anaphylaxis in mouse models (e.g., induced by histamine) [83].

Several mast-cell-released mediators are involved in inflammation but also in the activation of sensory nerve fibers to induce itch (histamine, serotonin, endothelin, IL-31, and proteases). In this context, it is noteworthy that mast cells are localized in close proximity to afferents innervating the skin. The activation of the IgE-receptor-mediated formation of S1P might thus also participate in the induction of itch (see below).

### 3.5. Sphingosine 1-Phosphate and T-Lymphocytes of the Skin

Human skin is home to abundant populations of αβ as well as γδ T cell receptor (TCR)-expressing T cells. Upon microbial invasion, APCs migrate from the skin to the lymph node resulting in the generation of effector αβ TCR-expressing T cells that travel to the site of inflammation. These effector T cells clear the infection typically within 3–10 days. Upon microbial eradication, the majority of effector T cells die, while subsets of αβ TCR-expressing T cells become recirculating memory cells or remain in the skin as tissue-resident memory T cells. Recirculating memory T cells exit in response to gradients of S1P and the chemokine ligand 21 (CCL21) via S1PR1 and the chemokine receptor CCR7. In contrast to recirculating memory T cells, tissue-resident memory T cells are often characterized by the expression of CD69. Indeed, CD69 is able to form a complex with S1PR1. Following the formation of this complex, S1PR1 is internalized and degraded within the cell, inhibiting its ability to bind S1P and initiate downstream signaling. This mechanism contributes to the maintenance of cells in the skin.

However, a unique T cell subgroup has been identified in the dermis, which is characterized by the expression of a γδ heterodimeric TCR on the cell surface. In contrast to αβ T cells, γδ T cells are non-MHC-restricted in recognizing antigens and, therefore, may have an innate immune signaling function [84,85,86]. Under physiological conditions, γδ T cells largely reside in the dermis and migrate only to a small extent to the draining lymph nodes in the skin. It has been indicated that dermal γδ T cells are a major origin of IL-17 secretion in response to skin infections. Thus, this T cell subtype possesses a crucial role in neutrophil recruitment and, therefore, a primary defense against pathogens [86,87]. Moreover, an increased turnover of γδ T cells from the dermis to the lymph nodes takes place under inflammatory conditions. Recently, a specific role of the S1PR2 has been identified in γδ T cell migration [88]. Thus, γδ T cells lacking both S1PR2 and CD69 show an increased migration rate from the dermis to the lymph nodes indicating a specific role of the S1PR2 in restraining the egress of these tissue-resident cells. In this context, it is of interest that the acute depletion of γδ T cells is protective in an animal model of psoriasis [89]. Thus, the topical administration of S1PR2 modulators may affect the progression of diseases such as psoriasis, which are characterized by a substantial immune response where tissue-resident lymphocytes are involved.

## 4. Sphingosine 1-Phosphate and Inflammatory Skin Diseases

### 4.1. Sphingosine 1-Phosphate and Psoriasis

Psoriasis is a chronic, immune-mediated, multifactorial skin disease that impacts 2–3% of the population worldwide. Classically, clinical symptoms are well-demarcated erythematous plaques with silvery scales. These plaques are usually found on the scalp, extensor surfaces and buttocks but can occur at any place on the body.

The histologic features are epidermal hyperplasia, a loss of the granular layer (hypogranulosis), a thickening of the stratum corneum (hyperkeratosis), and an incomplete cornification (parakeratosis). Moreover, the elongation and fusion of the rete ridges, dilated, protruding blood vessels in the dermis, and an inflammatory infiltrate of T cells and neutrophils (Munro’s abscess) are present in psoriasis [90]. The thickened epidermis is a consequence of premature hyperproliferating keratinocytes and incomplete cornification. Psoriasis is attributed to multiple interactions between keratinocytes and immune cells, especially Th1 and Th17 cells [91]. It can be assumed that the inflammatory cascade starts with skin antigens stimulating neutrophils and DCs, resulting in the release of cytokines, such as TNFα, IL-23, and IL-17. Thus, it is not astonishing that a systemic action of S1PR1 receptor modulators seems to be beneficial in the treatment of psoriasis. Ponesimod, a selective modulator of S1PR1, S1PR4, and S1PR5, has been tested in a phase 2 study in psoriasis. This 28-week, randomized, double-blind, placebo-controlled study investigated the efficacy, safety, and tolerability of treatment with ponesimod versus placebo in adult patients with moderate-to-severe chronic plaque psoriasis. Although clinical data indicate a significant reduction in psoriasis area and severity Index (PASI75), ponesimod is no longer being studied in this patient population. A reason could be adverse events, which include dyspnea, elevated liver enzymes, headache, nasopharyngitis, dizziness, bradycardia, pruritus, and cough [45].

Both environmental and genetic factors are involved in the pathophysiology of psoriasis. Common environmental triggers of psoriasis involve bacterial and viral infections, stress, smoking, obesity, and probably diet [92,93]. Therefore, it is not surprising that several studies indicate an altered plasma lipid homeostasis, including S1P, in psoriasis patients. It has been well established that the plasma S1P content is higher in psoriatic patients. However, studies are not entirely consistent as to whether S1P levels correlate with disease severity [94,95,96]. Similarly, there are also contradictory results if S1P levels return to lower levels when psoriasis is successfully treated [94,97]. Thus, it is not yet clear whether increased levels of S1P contribute to the pathogenesis of the disease or represent an outcome of the multifaceted skin disease. Therefore, several psoriasis animal studies have been performed to investigate the role of S1P within the skin in this inflammatory disease.

Most recently, a mouse line carrying a missense mutation of the S1P lyase was investigated under inflammatory conditions [98]. This mutation is accompanied by a dysfunctional S1P lyase activity leading to increased S1P contents within the skin. The mutant mice do not show visible skin alterations under steady-state conditions, as their ear anatomy and thickness are similar to wild-type mice. However, upon treatment with imiquimod, the mutant mice displayed exacerbated skin inflammation as manifested by increased acanthosis and orthokeratotic hyperkeratosis. Imiquimod as a TLR7 agonist is widely used as a psoriasis model in mice. Mechanistically, imiquimod activates DCs and other innate immune cells, which recruit and prime IL-17-producing γδ T cells within the treated skin. Most interestingly, S1P lyase mutant mice showed an increased number of γδ T cells in the skin. This is consistent with the fact that the conditional depletion of γδ T cells, but not constitutive deficiency, is connected with protection from imiquimod-induced psoriasis [89]. While these results suggest that the elevation of S1P via dysfunctional S1P lyase levels contributes to the progression of psoriasis, another study shows the opposite entirely. Jeon et al. indicated that the inhibition of S1P lyase improves psoriasis in the imiquimod-mediated psoriasis model [53]. Thus, the subcutaneous administration of an S1P lyase inhibitor resulted in diminished symptoms of psoriasis, such as erythema, scaling, and epidermal thickness in mouse skin due to a decreased proliferation and increased differentiation of keratinocytes. However, there was no effect on the severity of inflammation after treatment with the S1P lyase inhibitor.

The direct topical administration of S1P has also been investigated in the imiquimod-induced psoriasis mouse model [99]. The results of this study revealed that topical administration of S1P decreases the inflammatory reaction of the ears induced by imiquimod. Although Th17 cells and IL-17 secretion play a crucial role in maintaining inflammation in psoriatic skin, topical treatment of S1P showed no effect on this cytokine. This seems to be in accordance with the study of Liao et al., who indicated S1P possesses the ability to rather enhance the development of Th17 cells [100]. However, a pronounced attenuated epidermal hyperproliferation could be detected in the imiquimod model after topical treatment with S1P. This effect was confirmed in the severe combined immunodeficiency mice (SCID) model, where S1P-treated skin from patients suffering from psoriasis showed a decrease in epidermal thickness compared to the vehicle [99]. It is of interest that FTY720 did not show such an effect on hyperproliferation, which is consistent with the fact that the S1P-induced inhibition of keratinocytes is mediated via the S1PR2 receptor subtype [59].

While these studies indicate a beneficial effect of the topical application of S1P on psoriasis, further studies exist demonstrating that inhibition of SPHK2 leads to an improvement of psoriasis. Shin et al. figured out that the topical application of SPHK2 inhibitors, namely ABC294640 and HWG-35D, alleviated imiquimod-induced skin lesions and reduced the serum IL-17 levels induced by the application of imiquimod [55,56]. Moreover, the application of these inhibitors also decreased the skin mRNA levels of the genes associated with inflammation and keratinocyte differentiation elevated by imiquimod. These data suggest that the inhibition of SPHK2 possesses a protective effect in the imiquimod psoriasis model. However, it should be mentioned that the inhibitors of SPHK1 are not effective [55]. Rather, further studies indicated that at least ABC294640 has further effects on enzymes of the sphingolipid metabolism, which even leads to unexpected increased levels of S1P levels in different cells [57,58]. Therefore, it would be of great interest to investigate the influence of SPHK2 inhibitors on the S1P contents in keratinocytes.

Taken together, it seems certain that altered S1P levels of the skin have a significant influence on psoriasis, modulating skin and immune cells in a divergent manner. While elevated S1P levels are mainly able to inhibit the proliferation of keratinocytes and promote their differentiation, elevated S1P levels seem to promote a Th17 response. If these effects are due to different mechanisms or different S1P receptor subtypes, S1P modulators can be considered novel therapeutic approaches for the treatment of psoriasis.

### 4.2. Sphingosine 1-Phosphate and Atopic Dermatitis

Atopic dermatitis (AD) is a chronic inflammatory skin disease that typically occurs in early childhood. It is hallmarked by dry, erythematous, and itchy rashes. These rashes result in a vicious circle of itching and scratching, which is associated with a drastic deterioration in the quality of life [101]. It has been well-established that both environmental and genetic factors contribute to the development of AD [102]. Environmental factors include cold and dry weather, dampness, and more specific triggers such as house dust mites, pet fur, pollen, and molds. The best-known genetic cause leading to AD is a loss-of-function mutation in the FLG gene, which encodes for the structural protein filaggrin [103]. Additionally, a deficiency in cholesterol, fatty acid, ceramides, and tight junctions is visible in AD [104,105]. Consequently, penetration of allergens through the skin is strengthened, allowing them to interact with skin cells. Keratinocytes and dendritic cells act as important sentinels of the skin by recognizing danger signals or microbial pathogens and eliciting downstream immune responses, which include the production of type 2 cytokines. In addition to type 2 cytokines, type 17 cytokines have also been linked to the pathogenesis of AD [106].

S1P has also been shown to have a central role in AD, as the lipid mediator not only influences cytokine modulation but is also involved in the formation of the epidermal barrier. S1P serum levels have recently been found to be significantly enhanced in patients suffering from AD, and the levels of S1P correlate with the severity of the disease [107]. In atopic skin, it seems to be exactly the opposite, as an aberrant S1P metabolism has been shown in lesions of AD. Several studies indicate an increased S1P lyase activity in human and canine atopic lesions [108,109,110]. Thus, the levels of S1P in lesional canine skin are decreased compared to healthy controls [108]. These findings are in agreement with enhanced S1P lyase mRNA levels of lesional human and canine skin [109,110]. The diminished S1P levels suggest that topical application with S1P may be advantageous for the treatment of AD. Contact hypersensitivity is a well-established animal model to investigate immunological mechanisms of AD. In fact, in this model, the topical administration of S1P decreased the inflammatory reaction in the sensitization as well as in the elicitation phase [111]. S1P was able to decrease the weight and cell count of the draining auricular lymph node and reduce the inflammatory reaction.

Regarding the S1P receptor subtypes, S1PR2 has been identified as a crucial receptor subtype for skin barrier function (Figure 4) [60]. Although the skin of S1PR2-deficient mice exhibits no difference in phenotype and barrier function compared with that of wild-type mice, a higher TEWL after tape removal occurs in the S1PR2-deficient mice, indicating that S1PR2 is beneficial for epidermal barrier formation. All the more surprising are the studies in an animal model of AD, as the inflammatory response is significantly ameliorated in S1PR2-deficient mice [64,112]. Similarly, the topical administration of the S1PR2 antagonist JTE-013 is also able to suppress atopic responses in the ears and lymph nodes of wild-type mice. It has been suggested that this in vivo anti-atopic effect of topical JTE-013 treatment may be caused by the inhibition of S1PR2 in mast cells and DCs because S1PR2 has been proven to function as an activator of mast cell degranulation and DC maturation and migration [63,81,113,114,115]. Indeed, further studies are definitely needed to elucidate the important role of S1P and its signaling pathways in AD.

### 4.3. Sphingosine 1-Phosphate and Itch

Pruritus or itch is one of the main symptoms of AD and psoriasis. Itch evokes a desire to scratch and thus induces superficial damage to the skin. In the case of AD, this can lead to the vicious circle of further barrier disruption, leading to further allergen penetration and inflammatory exacerbation with increased itch sensation. There seems to be some overlap but also some striking differences concerning the signals inducing pain or itch that are initiated at primary sensory neurons in the skin by a wide variety of mediators or stimuli [116].

The sensory transduction of itch is realized by exciting a subpopulation of cutaneous, bare nerve endings (C-fibres and Aδ-fibres). It is questionable whether itch-specific nerve fibers exist. The current understanding is that these are rather a subspecies of nociceptive fibers. The cell bodies of these fibers are found in the dorsal root ganglia (DRG). The knowledge about excitation and signal transduction from the DRG to the spinal cord and brain has increased dramatically within the last decade but is still far from completely understood. Single-cell RNA sequencing experiments using mouse DRG revealed a diverse population of DRG cells that can be grouped into 11 distinctive clusters. Among those, itch-associated markers, such as histamine receptors, serotonin receptors, endothelin receptor A, or IL-31 receptor A, were quite selectively enriched in three subtypes of itch neurons termed NP1, NP2, and NP3. Histomorphologically, these represent non-myelinated, small-diameter neurons that also respond to a variety of itch-inducing substances in vitro [117]. A recent study on human DRG confirmed several of the findings in mice but also displayed some striking differences [118]. After the activation of itch-related receptors on DRG cells by, e.g., IL-31 or histamine, intracellular pathways activate specific ion channels, such as transient receptor potential vanilloid 1 (TRPV1) or ankyrin 1 (TPRA1), which finally evokes the discharge of the neurons [116]. The initial itch signal is transmitted from the periphery to the DRG and into the dorsal horn of the spinal cord. The ascending signals are then transmitted through the lateral spinothalamic tract into the brain, where several itch-processing areas have been identified, e.g., thalamic nuclei, part of the cortex, amygdala, and nucleus accumbens. The descending signals include signals from the areas that belong to the reward system, such as the periaqueductal gray and the striatum, and are manifested in scratching behavior [119].

The itch associated with inflammatory skin diseases, such as AD and psoriasis, is triggered by certain endogenous cytokines, such as IL-31 and TSLP, as well as biogenic amines, such as histamine and serotonin. However, lipids, such as leukotriene B4, and sphingolipids, such as sphingosylphosphorylcholine (SPC), have been associated with itch [120]. The possible involvement of S1P in the mediation of pain was elucidated quite early, and recent publications indicate that S1P is also associated with itch, although only very few studies exist and with partly contradictory results. S1P activates S1PR3, which induces both itch and pain. Thus, it has been suggested that itch transduction is due to the activation of TRPA1 via the Gβγ signaling pathway, but pain transduction is realized by TRPV1 activation via PLC-mediated signal transduction [121]. On the contrary, it has been discovered that itch-related scratching is decreased in TRPV1-deficient mice but not in TRPA1-deficient mice [122]. Interestingly, in the latter work, the selective S1PR2 agonist CYM 5520 induces a distinct calcium signal in sensory neurons, indicating its role in sensory perception (Figure 4). In addition, it has been demonstrated that S1PR3-deficient mice show less itch behavior in the imiquimod-induced psoriasis model but not in an AD-like model (induced by MC903) [123]. However, further studies are urgently needed to elucidate the exact role of S1P in acute and chronic itch associated with AD and psoriasis.

## 5. Conclusions

Recent research clearly indicates the role of S1P as a pivotal signaling lipid in inflammatory and pruritic skin conditions, such as atopic dermatitis and psoriasis. Particularly, subtype-specific agonists and antagonists help to further understand the complex role of S1P signaling, and it seems promising to further elucidate these roles, especially the role of S1PR2 and S1PR3 in inflammatory skin diseases and itch. This is of great interest as interfering with S1P signaling pathways may represent an innovative option for the treatment of inflammatory skin diseases and itch.

## Figures and Tables

**Figure 1 ijms-24-01456-f001:**
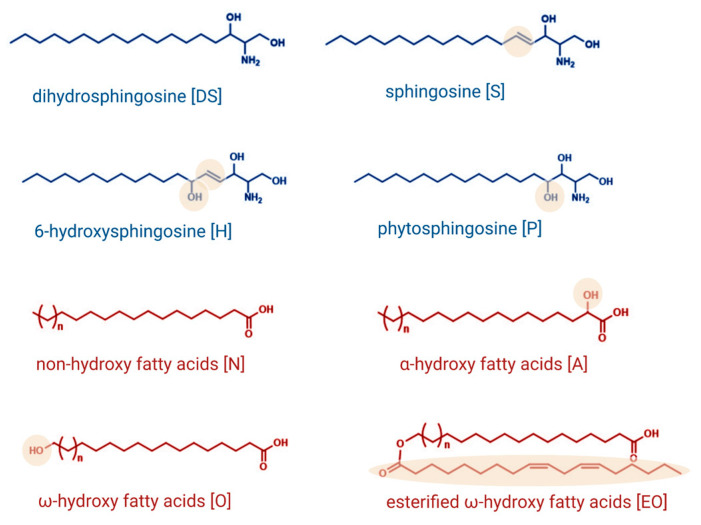
Structures and nomenclature of sphingoid backbones and fatty acids that have been identified to occur at the sphingoid base. Created with BioRender.com.

**Figure 2 ijms-24-01456-f002:**
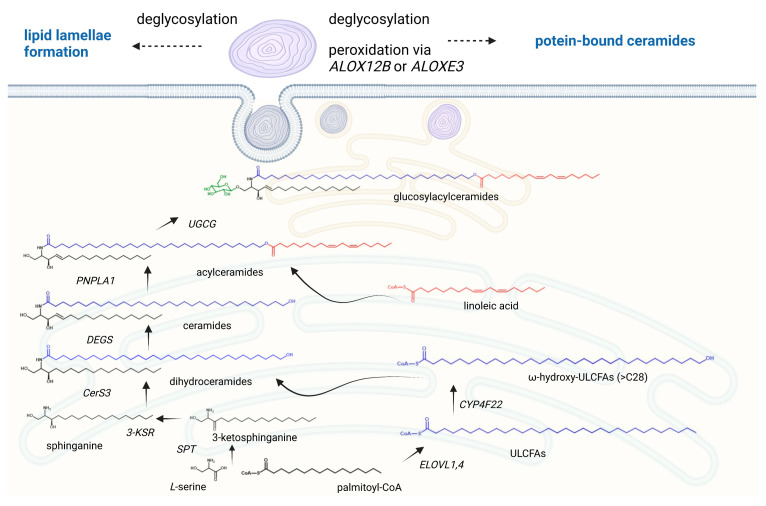
The formation of acylceramides in the human epidermis. Acylceramides are formed in the endoplasmic reticulum and secreted into the extracellular environment through the Golgi apparatus and assembling of lamellar bodies. The de novo pathway starts with the condensation of the serine and palmitoyl-CoA via the serine palmitoyl transferase (SPT). The resultant 3-ketosphinganine is then reduced by 3-ketosphinganine reductase (3-KSR) to sphinganine. The fatty acid elongases ELOVL1 and ELOVL4 yield ultralong-chain fatty acids of up to C26 and C28 carbon-chain-lengths, which are hydroxylated via CYP4F22 at the ω-position. These generated ω-hydroxylated ultralong-chain fatty acids (ULCFAs) are linked to the amino group of sphinganine by the ceramide synthase 3 (CerS3), leading to the formation of dihydroceramides. The introduction of a double bond between carbons C-4 and C-5, mediated by dihydroceramide desaturase, forms the central product of ceramide. The transacylase PNPLA1 (Patatin-like Phospholipase Domain Containing 1) generates an ester linkage between the fatty acid taken from triglycerides and the ω-hydroxy group from ceramide to create the acylceramide. In the Golgi apparatus, acylceramides are glycosylated by the UDP-glucose ceramide glucosyltransferase UGCG, followed by an ABCA12-mediated transport into lamellar bodies. After secretion, glucosylacylceramides are reconverted back to their ceramide species to form organized lipid lamellae. For the generation of protein-bound ceramides, the fatty acid portions of acylceramides are oxidized via the lipoxygenases ALOX12B and ALOXE3, which enables a cross-link of the ω-hydroxyl group with cornified envelope proteins such as involucrin and envoplakin. Created with BioRender.com.

**Figure 3 ijms-24-01456-f003:**
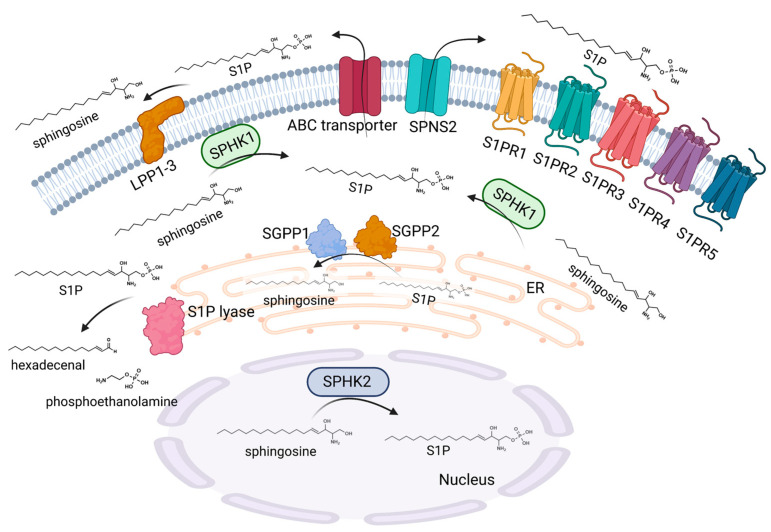
S1P metabolism and signaling. S1P is synthesized via SPHK1 and SPHK2. SPHK2 is mainly located in the nucleus, whereas SPHK1 is present in the cytoplasm. SPHK1 can translocate to the plasma membrane after stimulation, which facilitates the secretion of S1P into the extracellular environment. SPNS2 and several ABC transporters have been identified as transporters of S1P through the cell membrane. Extracellular S1P can act via five G-protein coupled receptors, namely S1PR1-5. The degradation of extracellular S1P occurs via three lipid phosphate phosphatases, LPP1-3. Intracellular S1P is metabolized by two phosphatases, SGPP1 and 2, which are located at the endoplasmic reticulum membrane with the active site on the luminal side. An irreversible degradation of S1P is catalyzed by the S1P lyase, which is also present in the endoplasmic reticulum with the active site on the cytosolic side, leading to the formation of hexadecenal and phosphoethanolamine. Created with BioRender.com.

**Figure 4 ijms-24-01456-f004:**
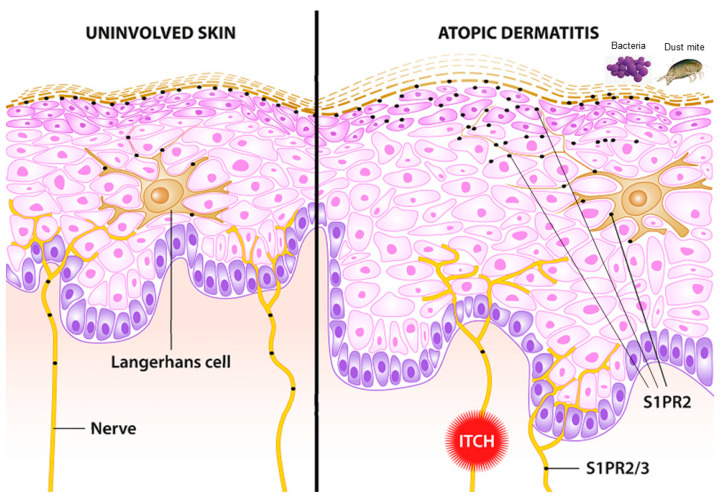
Schematic of uninvolved skin (**left**) and lesional skin in atopic dermatitis (**right**). S1PR2 and partially S1PR3 are involved in crucial pathways associated with atopic dermatitis. Lichenification and skin proliferation are under the control of S1PR2. Bacterial infection with *S. aureus* induces an alarmin function in keratinocytes mediated via S1PR2. Langerhans cells are pivotal antigen-presenting cells in the skin. The uptake of antigen and allergens is modulated via S1PR2. Itch, a pivotal symptom of atopic dermatitis, is partially mediated via S1PR2 and S1PR3.

## Data Availability

Not applicable.

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
