# Peer review of "Sphingosine 1-Phosphate as Essential Signaling Molecule in Inflammatory Skin Diseases"

_ijms, 2023, doi:10.3390/ijms24021456_

Round 1

Reviewer 1 Report

This is a well-written review about the role of sphingosine 1-phosphate in inflammatory skin diseases. The authors presented an overview of sphingosine 1-phosphate signaling pathways in various skin and immune cells. In summary, this is a very instructive review, which deserves publication in IJMS.

Author Response

We are delighted that the reviewer found the manuscript of great interest.

Reviewer 2 Report

Comments and suggestions for Authors

This is an exciting review of S1P and its roles on skin resident cells and skin diseases. This review describes S1P well from a molecular point of view and in clinical contexts.

I am attaching a few suggestions that might further improve the quality of this review:

Major comments:

1.    The authors focus on ceramide in the introduction. However, considering that the content of the later sections is the relationship between S1P and various cells and skin diseases, the authors should mainly focus on S1P in the introduction instead of ceramides. For example, the content of 1.2. should be considered as an introduction, and the content of 1.1. should be moved to a later chapter with more attention to describing the relationship between S1P and ceramides.

2.    Several repetitions can be removed (page 8, line 327-336 and page 14, line 618-626; page 8, line 370-374, and page 14, line 626-629).

Minor comments:

1.    Page 7, lines 252-268, describing S1P and psoriasis. This passage should be moved into 4.1.

2.    The histopathology of psoriasis is insufficiently described, as in 4.1., p12, lines 518-520. Loss of granular layer (hypogranulosis), thickening of stratum corneum (hyperkeratosis), incomplete cornification (parakeratosis), neutrophilic infiltration in the horny layer (Munro’s abscess), elongation and fusion of rete ridges are also histopathological findings quite characteristic of psoriasis. They correspond to the clinical manifestations of plaque and silvery scale described by the authors.

3.    In chapter 4.2., page 13, lines 593-594, please describe the environmental factors associated with atopic dermatitis.

4.    In figure 3, the authors should consider having nomenclature for sphingolipids instead of a chemical structural formula only. It would make the figure more visible and attractive.

Author Response

The reviewers' advice was of great interest and help to us and we have tried to implement this advice as well as possible. In fact, we find that this has improved the manuscript.

In detail, we have made the following changes.

The authors focus on ceramide in the introduction. However, considering that the content of the later sections is the relationship between S1P and various cells and skin diseases, the authors should mainly focus on S1P in the introduction instead of ceramides. For example, the content of 1.2. should be considered as an introduction, and the content of 1.1. should be moved to a later chapter with more attention to describing the relationship between S1P and ceramides.

We agree with the reviewer´s comment that at the beginning of the article the sphingolipids and their role in the formation of the epidermal barrier were described in great detail. We just wanted to make a clear distinction between the barrier function, structurally enabled by sphingolipids, and the role of sphingolipids as signaling molecules. Since this was not clearly expressed, we have again clearly distinguished this in the introduction. We hope that this is in agreement with the reviewer’s suggestion.

Several repetitions can be removed (page 8, line 327-336 and page 14, line 618-626; page 8, line 370-374, and page 14, line 626-629).

Indeed, the reviewer is right that there were repetitions in the function of the S1PR2 receptor. Thank you for the advice. We have removed those repetitions.

Page 7, lines 252-268, describing S1P and psoriasis. This passage should be moved into 4.1.

We agree with the reviewers comment that the S1P/ponesomid/psoriasis fits much better in section 4.1. We have incorporated this paragraph now in section 4.1  

The histopathology of psoriasis is insufficiently described, as in 4.1., p12, lines 518-520. Loss of granular layer (hypogranulosis), thickening of stratum corneum (hyperkeratosis), incomplete cornification (parakeratosis), neutrophilic infiltration in the horny layer (Munro’s abscess), elongation and fusion of rete ridges are also histopathological findings quite characteristic of psoriasis. They correspond to the clinical manifestations of plaque and silvery scale described by the authors.

Thank you for this helpful advice to describe the histopathology in more detail. We have now included this in the psoriasis section.

In chapter 4.2., page 13, lines 593-594, please describe the environmental factors associated with atopic dermatitis.

We now have included the environmental factors that triggers atopic dermatitis. This fits very well as also the genetic factors have been described.

In figure 3, the authors should consider having nomenclature for sphingolipids instead of a chemical structural formula only. It would make the figure more visible and attractive.

This is a really good idea. We have left the structures as they are of interest for some readers, but have also used now the nomenclature.

We hope that these changes are in line with the reviewers' suggestions.

Reviewer 3 Report

The authors in this review described the involvement of the sphingolipid sphingosine 1-phosphate (S1P) in processes such as proliferation and differentiation of keratinocytes.  Moreover, it is increasing the knowledge regarding an immunomodulatory role of S1P and the dysregulation in the signaling pathways of S1P is involved in pathophysiological conditions of skin diseases.  The authors in the present review, described and discussed the fundamental role of S1P in skin cells and in inflammatory skin diseases.  Kleuser and Bäumer conclude that the regulation of S1P signaling pathways may represent an innovative option for the treatment of inflammatory skin diseases.

In this reviewer opinion, the review is interesting and innovative.  The authors described with a complete overview on the role of S1P in skin cells and in inflammatory skin diseases.  For these reason, in this reviewer opinion, the review can be accepted for publication in the present form

Author Response

We are delighted that the reviewer found the manuscript innovative and interesting.